# Associations of time of day with cardiovascular disease risk factors measured in older men: results from the British Regional Heart Study

Claudio Sartini,[1] Peter H Whincup,[2] S Goya Wannamethee,[1] Barbara J Jefferis,[1] Lucy Lennon,[1] Gordon DO Lowe,[3] Paul Welsh,[3] Naveed Sattar,[3] Richard W Morris[4]

[1]Department of Primary Care & Population Health, University College London, London, UK
[2]Population Health Research Institute, St George's University of London, London, UK
[3]University of Glasgow, Institute of Cardiovascular and Medical Sciences, Glasgow, UK
[4]Centre for Academic Primary Care, Schoolof Social and Community Medicine, University of Bristol, Bristol, UK

**Correspondence to**
Claudio Sartini;
c.sartini@ucl.ac.uk

## ABSTRACT

**Objective** We estimated associations of time of day with cardiovascular disease (CVD) risk factors measured in older men.

**Methods** CVD risk factors (markers of inflammation and haemostasis, and cardiac markers) were measured on one occasion between 08:00 and 19:00 hours in 4252 men aged 60–79 years from the British Regional Heart Study. Linear models were used to estimate associations between time of day and risk factors. When an association was found, we examined whether the relationship between risk factors and cardiovascular mortality was affected by the adjustment for time of day using survival analyses.

**Results** N-terminal pro-brain natriuretic peptide (NT-proBNP) levels increased by 3.3% per hour (95% CI 1.9% to 4.8%), interleukin-6 (IL-6) increased by 2.6% per hour (95% CI 1.8% to 3.4%), while tissue plasminogen activator (t-PA) decreased by 3.3% per hour (95% CI 3.7% to 2.9%); these associations were unaffected by adjustment for possible confounding factors. The percentages of variation in these risk factors attributable to time of day were less than 2%. In survival analyses, the association of IL-6, NT-proBNP and t-PA with cardiovascular mortality was not affected by the adjustment for time of day. C reactive protein, fibrinogen, D-dimer, von Willebrand factor and cardiac troponin T showed no associations with time of day.

**Conclusions** In older men, markers of inflammation (IL-6), haemostasis (t-PA) and a cardiac marker (NT-proBNP) varied by time of day. The contribution of time of day to variations in these markers was small and did not appear to be relevant for the CVD risk prediction.

## BACKGROUND

Previous studies have reported time of day variation in both established and emerging cardiovascular disease (CVD) risk factors in middle-aged adults, such as blood pressure, lipids and some well-established inflammatory and haemostatic factors (eg, white cell, red blood cell and platelet counts).[1–3] However, the extent to which some emerging CVD risk factors such as interleukin (IL)-6 , a marker of inflammation causally associated

with CHD in a recent study,[4] and N-terminal pro-brain natriuretic peptide (NT-proBNP), a marker of heart failure,[5] vary by time of day have been less studied. Moreover, very little is known on time of day variations in other emerging risk factors prospectively associated with CVD (eg, tissue plasminogen activator (t-PA), D-dimer, von Willebrand factor (vWF) and cardiac troponin T (cTnT)), although their causal association with CVD remain debated or not yet tested.

We would expect that time of day variations in some emerging CVD risk factors measured in older adults may occur, consistent with findings in younger populations.[6] However, in older adults, the degree of difference attributable to time of day has not been yet estimated; establishing its importance and its effects on prediction of CVD risk is important given the potentially wider use of NT-proBNP in risk stratification (as shown in a recent major meta-analysis in the general population[7]) and potential causal link between IL-6 and CVD.[4] Therefore, the aim of this study was to investigate how emerging CVD risk factors, including markers of inflammation, haemostasis and myocardial function, varied by time of day in older British men.

## METHODS

### Participants

The British Regional Heart Study (BRHS) is a prospective cohort study of CVD involving 7735 middle-aged men (40–59 years) selected in 1978–1980 from the age–sex registers of one local primary care centre in 24 British towns.[8] The 24 towns were selected to represent the variation in CVD across the UK.[9] Participants provided informed written consent to the investigation, which was performed in accordance with the Declaration of Helsinki.[10]

### Follow-up examination

In 1998–2000, an average of 20 years after the initial recruitment, 4252 surviving participants (77% response rate) aged 60–79 years who were resident in the UK attended a physical examination during which nurses took a fasting blood sample on one occasion for each participant. The men were asked to fast for a minimum of 6 hours, during which they were instructed to drink only water, as previously reported.[2] The blood samples were collected between 08:00 and 19:00 hours and then assayed for a range of biochemical and haematological markers. Participants' appointment times were non-systematically allocated. They were offered the opportunity to contact the BRHS team and change the time of examination if unable to attend; a small proportion of participants did so.

The participants were also asked to complete a questionnaire that included questions on other established CVD risk factors, such as age, social class, smoking habits, alcohol consumption and physical activity. Specifically, physical activity levels were self-reported[11] and recently validated using accelerometers.[12] Incident CVDs, including non-fatal stroke and non-fatal myocardial infarction (MI), were recorded: their definitions have been reported elsewhere.[13] Men were also asked whether a doctor had ever told them that they had heart failure.[5] The number of blood samples collected and included in the analyses differ according to the risk factor measurements (the number of observations varied from 3580 for NT-proBNP to 3863 for von Willebrand Factor in complete case analyses including all covariates of interest).

### CVD risk factors

Circulating levels of markers of inflammation (C reactive protein (CRP), IL-6 and fibrinogen), cardiac markers (NT-proBNP and cTnT) and markers of haemostasis (t-PA antigen, fibrin D-dimer and vWF) were measured.

D-dimer and t-PA levels were measured using an ELISA (Biopool AB, Umeå, Sweden), as was vWF antigen (Dako, High Wycombe, UK). CRP was assayed using ultrasensitive nephelometry (Dade Behring, Milton Keynes, UK). IL-6 was assayed using a high-sensitivity ELISA (R&D Systems, Oxford, UK). Fibrinogen was assayed using an automated Clauss assay in a coagulometer (MDA-180, Organon Teknika, Cambridge, UK). NT-proBNP and cardiac troponin T (cTnT) were measured in plasma

samples on an automated clinically validated immunoassay analyser (e411, Roche Diagnostics, Burgess Hill, UK) using the manufacturers' calibrators and quality control reagents. Intra-assay and interassay coefficient of variations were respectively: 4.1% and 6.6% for t-PA; 3.2% and 4.2% for vWF; 4.7% and 5.2% for D-dimer; 4.7% and 8.3% for CRP; 7.5% and 8.9% for IL-6; 2.6% and 3.7% for fibrinogen and 4.4% and 7.7% for NT-proBNP and cTnT.

The samples were centrifuged and separated on the morning or afternoon of collection and stored on site at −20°C until they were transferred to a central freezer storage location at −70°C within 2 weeks of sample collection. Samples were then transferred on dry ice to a single central laboratory and were thawed immediately before analysis. Plasma samples were used for all the analyses reported here. The original sample collection took place between January 1998 and March 2000. Most of the analyses described here were carried out during 2000, after a maximum of 3 years storage; NT-proBNP and cTNT were analysed in 2009.

### Statistical methods

First, the distributions of the outcomes were examined; the outcomes were log-transformed as the distributions were positively skewed. Therefore, analysis was carried out on their log-transformed values throughout. Unadjusted geometric means and 95% CIs of the outcomes were plotted against hour of the day.

#### Adjusted associations between time of day and the outcomes

Associations between time of day (fitted as a continuous variable, range 8–18) and the outcomes were examined using linear multilevel random intercept models (level 1=individual, level 2=town of residence). The results can be interpreted as between-person variations over the course of the examination day; the estimates from the linear model were reported as the difference in the outcome levels per hour of sampling over the examination day. As the outcomes were log-transformed, the results were reported as per cent difference in the outcome geometric mean per hour of sampling. All models were initially adjusted for age only. Next, the models were adjusted for age and other possible confounding factors: social class, body mass index, previous stroke or MI, physical activity, smoking status, alcohol consumption, use of statin and a seasonal term (fitted using a cosinor function, as in previous studies).[14] As NT-proBNP and cTnT are principally markers of heart failure, the association with time of day was adjusted for previous heart failure.

When the association of time of the day with the outcomes was found to be statistically significant, the proportion of variance associated with time of the day was estimated using partial $R^2$.

#### Sensitivity analyses

Six sensitivity analyses were performed: (1) all models were additionally adjusted for fasting time and diabetes; (2) all models were carried out excluding men with

diabetes; (3) interactions were fitted to test whether the time of day associations were modified by age (fitted as continuous variable); (4) as NT-proBNP and cTnT were acknowledged as specific cardiac markers,[5] interactions were fitted to test whether the time of day associations were modified by previous heart failure (yes/no); (5) to explore the potential of undiagnosed heart failure or cardiac damage influencing findings for NT-proBNP and cTnT, we repeated regression models after excluding men with NT-proBNP >400 pg/mL; and (6) a quadratic term for time of day was added to the models in order to check for non-linearity.

As IL-6 has been causally associated with cardiovascular risk,[4] and prospectively associated with CVD mortality in the BRHS sample used here,[15] we investigated the relevance of time of day to the cardiovascular risk prediction by performing two survival analyses: in the first analysis, we used Cox models where unadjusted log IL-6 was used as the predictor and CVD mortality as the clinical outcome, then we repeated the same analysis using log IL-6 standardised by the time of day rather than unadjusted log IL-6. For completeness of information, we repeated this sensitivity analysis for NT-proBNP and t-PA.

## RESULTS

The characteristics of the study participants (mean age 68.7 years, SD=5.5) are reported in table 1. The associations between time of day (by hour) and risk factors are shown in figure 1. Evidence of an increase over the course of the day was particularly noticeable for IL-6 and for NT-proBNP (figure 1). Also, levels of t-PA were lower in the afternoon in comparison with morning, while variations by time of day for other risk factors were not clearly observable from the plots (figure 1). The results of corresponding linear regression analyses are shown in table 2; statistically significant associations between time of the day and some outcomes were found (table 2); over the course of the examination day, NT-proBNP levels increased by 3.3% per hour (95% CI 1.9% to 4.8%) and IL-6 increased by 2.6% per hour (95% CI 1.8% to 3.4%). Conversely, t-PA decreased by 3.3% per hour (95% CI 3.7% to 2.9%). The proportion of variance associated with time of the day from the fully adjusted models was 0.5%, 1%, and 2% for NT-proBNP, IL-6 and t-PA, respectively.

### Sensitivity analyses

Overall, we found that fasting time did not alter the magnitude of associations between time of the day and the outcomes reported in table 2. Only the association between time of the day and t-PA was strongly attenuated after accounting for fasting time (fitted as continuous variable): the decrease in t-PA levels was −3.3% (95% CI −3.7 to −2.9) per hour before the adjustment (table 2) and −1.4% (95% CI −2.2% to −0.1%) after the adjustment for fasting. An additional adjustment for diabetes status did not alter the magnitude of the association between hour of the day and the outcomes. We also performed the

**Table 1** Individual characteristics and risk factor levels in the British Regional Heart Study of men who have attended the examination in 1998–2000

| Demographic and background characteristics | |
|---|---|
| Age (years), mean (SD) | 68.7 (5.5) |
| Social class (manual) | |
| Manual, n (%) | 2166 (51.1) |
| Non-manual, n (%) | 1966 (46.3) |
| Armed Forces, n (%) | 112 (2.6) |
| Physical health | |
| Body mass index, mean (SD) | 26.9 (3.7) |
| Prevalence of stroke/myocardial infarction, n (%) | 153 (3.6) |
| Prevalence of heart failure, n (%) | 390 (9.2) |
| Diabetes, n (%) | 478 (11.2) |
| Behavioural factors | |
| Smoking | |
| Never, n (%) | 1233 (29.1) |
| Ex-smokers, n (%) | 2464 (58.0) |
| Smokers, n (%) | 548 (12.9) |
| Alcohol consumption | |
| None, n (%) | 431 (10.3) |
| Occasional/light, n (%)* | 2949 (70.5) |
| Moderate/heavy, n (%)† | 779 (18.6) |
| Physical activity level | |
| Inactive, n (%) | 471 (11.5) |
| Occasionally, n (%) | 957 (23.4) |
| Light, n (%) | 767 (18.7) |
| Moderate, n (%) | 591 (14.4) |
| Moderate vigorous, n (%) | 690 (16.8) |
| Vigorous, n (%) | 621 (15.1) |
| CVD risk factor, geometric mean (SD) | |
| CRP, mg/L | 1.74 (3.03) |
| IL-6, pg/mL | 2.46 (1.94) |
| Fibrinogen, g/L | 3.19 (1.25) |
| t-PA, ng/mL | 10.23 (1.50) |
| vWF, IU/dL | 132.41 (1.40) |
| D-dimer, ng/mL | 84.32 (2.32) |
| NT-proBNP, pg/mL | 101.50 (3.32) |
| cTnT, pg/mL | 12.07 (1.64) |

*≥1 and ≤15 units per week (1 unit is approximately one drink, such as one glass of wine).

†≥16 units per week (1 unit is approximately one drink, such as one glass of wine).

CRP, C reactive protein; cTnT, cardiac troponin T; IL-6, interleukin-6; NT-proBNP, N-terminal pro-brain natriuretic peptide; t-PA, tissue plasminogen activator; vWF, von Willebrand factor.

analysis excluding men with diabetes completely (table 2, model 3), but the association between time of day and the outcomes did not substantially change.

For all outcomes, we also did not find evidence for an interaction between of time of day with age (results not shown).

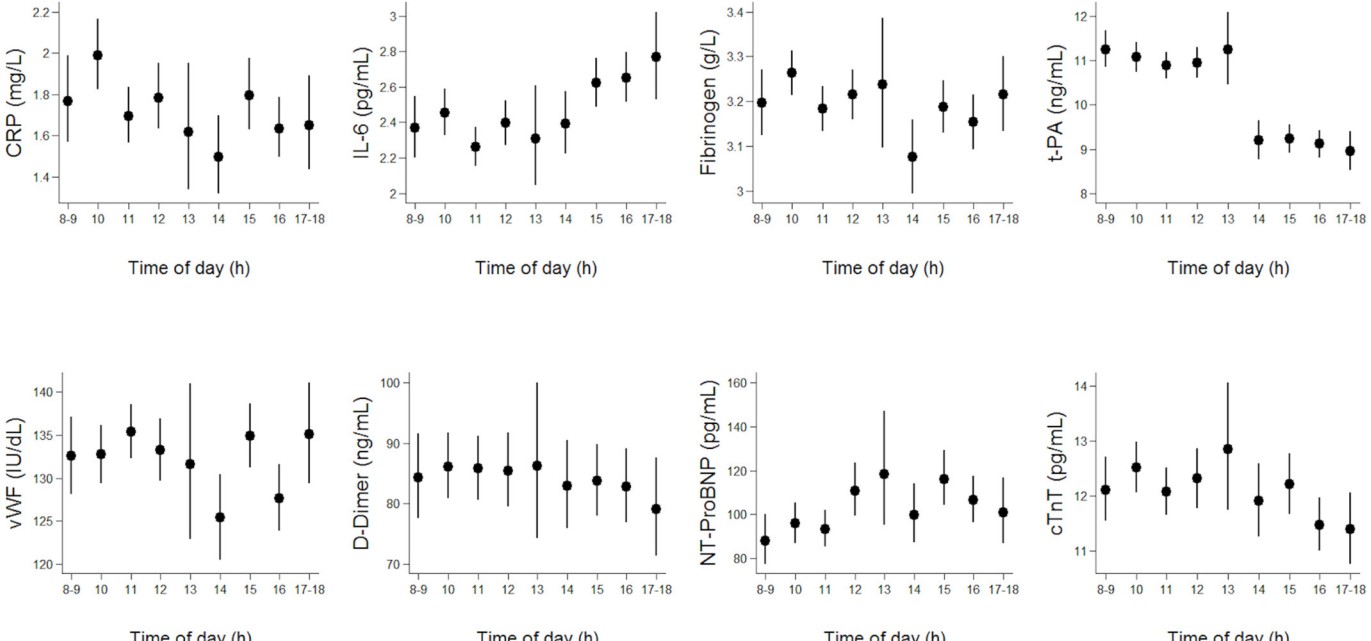

**Figure 1** Unadjsted geometric means (95% CI) by time of the day* for C reactive protein (CRP), interleukin-6 (IL-6), fibrinogen, tissue plasminogen activator (t-PA), von Willebrand factor (vWF), fibrin D-dimer, N-terminal pro-brain natriuretic peptide (NT-proBNP) and cardiac troponin T (cTnT) measured on one occasion in BRHS men aged 60–79 during the years 1998–2000. *Total number of men examined per hour was 33 (0.7%) at 08:00–08:59, 363 (8.5%) at 8:00–9:59 hours, 699 (16%) at 10:00–10:59 hours, 771 (18%) at 11:00–11:59 hours, 591 (14%) at 12:00–12:59 hours, 99 (2%) at 13:00–13:59 hours, 306 (7%) at 14:00–14:59, 560 (13%) at 15:00–15:59 hours, 566 (13%) at 16:00–16:59 hours, 260 (6%) at 17:00–17:59 and 3 (<0.1%) at 18:00–18:59.

In stratified analysis, NT-proBNP levels increased by 3.4% (95% CI 1.9% to 4.8%, p<0.001) per hour in older men without heart failure. Although men who previously had heart failure had increased NT-proBNP levels, there was no evidence for an interaction between previous heart failure with time of the day (p=0.954). After excluding 466 men with NT-proBNP level of >400 pg/mL (12% of the sample), associations between time of the day measures and NT-proBNP remained statistically significant and slightly increased in magnitude (3.9% (95% CI 2.7% to 5.1%), p<0.001). As reported in the main analysis, no significant associations were found between time of the day and cTnT in stratified analysis.

When adding a quadratic term to the model, we found a significant improvement in model fit for IL-6 only (p=0.030 for the time of day squared term). The association of time of day with IL-6 appeared to be slightly J-shaped, with a linear increase starting from 11:00 to 19:00 hours.

We examined whether adjustment for hour of day affected the associations between risk factors and CVD mortality. In survival analysis, higher levels of log IL-6 were associated with increased CVD mortality (HR=1.70, 95% CI 1.54 to 1.87). Standardising IL-6 by time of the day did not change the relationship (HR=1.71, 95% CI 1.55 to 1.88). Also, standardising NT-proBNP levels by time of the day did not alter the magnitude of the effect on CVD mortality (HR=1.92, 95% CI 1.81 to 2.04). Finally, associations of t-PA levels with increased CVD mortality did not change substantially before (HR=1.74, 95% CI 1.45 to 2.09) and after standardising (HR=1.77, 95% CI 1.47 to 2.14) by time of the day.

## DISCUSSION

To our knowledge, this is the largest investigation of relationships between time of day and CVD risk factors in older men. After adjusting our analysis for potential confounding factors, we demonstrated that some, but not all, CVD risk factors levels varied by time of day. In particular, NT-proBNP and IL-6 increased linearly over the course of the day. Conversely, a decrease in t-PA was also observed; however, after accounting for fasting time, the relationship with time of the day was strongly attenuated (therefore fasting time could partially explain the drop in t-PA levels observed in the afternoon vs morning). Our analyses showed that the contribution of time of the day to the overall variation of NT-proBNP, IL-6 and t-PA was small and without clinical importance; we observed that time of day did not have a sufficiently strong effect to be taken into account when assessing the impact of IL-6, NT-proBNP and t-PA on CVD mortality. Lastly, an association of time of day with other risk factors was not observed.

Literature on time of day variation in emerging CVD markers of inflammation and haemostasis in older adults is limited; to our knowledge, this is the first time these findings have been reported in older adults. Findings from earlier studies of younger adults were fairly consistent with ours. For example, a recent meta-analysis of several small studies that analysed IL-6 proposed a diurnal pattern, with

**Table 2** Cross-sectional adjusted associations between time of day (fitted as continuous variable) and cardiovascular disease (CVD) risk factors measured in the British Regional Heart Study (BRHS) men (aged 60–79) attending the follow-up year 20 examination in 1998–2000

| CVD risk factor‡ | Model 1: age adjusted* | | Model 2: fully adjusted† | | Model 3: fully adjusted† excluding men with diabetes | |
|---|---|---|---|---|---|---|
| | Per cent difference (95% CI) in the CVD risk factor levels per hour of sampling§ | p Value | Per cent difference (95% CI) in the CVD risk factor levels per hour of sampling§ | p Value | Per cent difference (95% CI) in the CVD risk factor levels per hour of sampling | p Value |
| NT-proBNP | **3.5 (2.0 to 5.0)** | **<0.001** | **3.3 (1.9 to 4.8)** | **<0.001** | **3.5 (2.0 to 5.0)** | **<0.001** |
| IL-6 | **2.6 (1.7 to 3.4)** | **<0.001** | **2.6 (1.8 to 3.4)** | **<0.001** | **2.4 (1.6 to 3.3)** | **<0.001** |
| t-PA | **−3.3 (−3.8 to −2.9)** | **<0.001** | **−3.3 (−3.7 to −2.9)** | **<0.001** | **−3.2 (−3.6 to −2.7)** | **<0.001** |
| Fibrinogen | −0.2 (−0.5 to 0.0) | 0.088 | −0.2 (−0.5 to 0.1) | 0.104 | −0.2 (−0.5 to 0.1) | 0.149 |
| cTnT | −0.4 (−0.9 to 0.2) | 0.194 | −0.4 (−1.0 to 0.2) | 0.174 | −0.4 (−1.0 to 0.2) | 0.165 |
| CRP | −1.0 (−2.3 to 0.4) | 0.151 | −0.9 (−2.2 to 0.4) | 0.175 | −0.9 (−2.2 to 0.5) | 0.191 |
| vWF | −0.2 (−0.6 to 0.2) | 0.374 | −0.2 (−0.6 to 0.2) | 0.380 | −0.1 (−0.5 to 0.3) | 0.703 |
| D-dimer | −0.0 (−1.0 to 1.0) | 0.929 | −0.1 (−1.0 to 0.9) | 0.890 | −0.1 (−1.2 to 0.9) | 0.801 |

Associations are reported as per cent difference in CVD risk factors levels per 1 hour of sampling over the examination day (08:00–19:00 hours). The statistically significant associations are marked in bold

*Model 1: two level linear models (level 1=person and level 2=town of residence during the BRHS recruitment) adjusted for age. Model 1 used the same number of observations of Model 2 (complete case analysis).

†Model 1 additionally adjusted for social class, body mass index, smoking status, alcohol consumption, physical activity, use of statin and season. Associations with IL-6, t-PA, fibrinogen, CRP, vWF and D-dimer were additionally adjusted for prevalence of stroke/MI, while association of time of the day with NT-proBNP and cTnT models were additionally adjusted for prevalence of heart failure.

‡Model 1 and Model 2 used the same number of observations: 3580 for NT-proBNP, 3832 for IL-6, 3863 for t-PA, 3861 for fibrinogen, 3827 for cTnT, 3838 for CRP, 3863 for vWF, 3859 for D-dimer.

§Model 3 number of observations: 3176 for NT-proBNP, 3398 for IL-6, 3429 for t-PA, 3427 for fibrinogen, 3398 for cTnT, 3405 for CRP, 3429 for vWF, 3425 for D-dimer.

CRP, C reactive protein; cTnT, cardiac troponin T; IL-6, interleukin-6; MI, myocardial infarction; NT-proBNP, N-terminal pro-brain natriuretic peptide; t-PA, tissue plasminogen activator; vWF, von Willebrand factor.

overall IL-6 levels increased between 08:00 and 19:00 hours as in our study.[16] However, in two earlier very small studies of 12[17] and 5[18] participants, IL-6 peaked in the night-time. It is possible that peaks in IL-6 levels may be associated with cognitive symptoms of depression[19] and daily activities, although in the BRHS population, this has not yet been investigated. One previous study found that BRHS men were more active in the morning and in early afternoon[20] when the main activities were usually gardening, house works, shopping or leisure walking. Whether IL-6 was implicated in this daily pattern remains uncertain and can potentially be explored in future studies.

Moreover, one previous study reported increased levels of NT-ProBNP over the course of day[21] as we observed in our study. A decrease in t-PA over the examination day was also reported in younger subjects (a 45-year-old UK population of 9377 men and women)[6]; however, t-PA did not vary by time of the day in a previous large study of 1288 healthy men and women aged 25–64 years.[22]

In comparison with our study, findings regarding CRP, fibrinogen, D-dimer, vWF and cTnT reported in earlier studies of younger adults were similar; a few previous studies reported that they did not find an association of time of day with CRP,[23] D-dimer[24] and vWF.[25] In one study, the variation in CRP, fibrinogen, D-dimer and vWF attributable to time of day was minimal.[6] Literature on cTnT is scarce; one small

previous study of repeated measures in seven participants with type 2 diabetes reported a decrease in cTnT between 08:00 and 20:00.[26]

Although one previous study suggested that diurnal variations in CVD risk factors could be relevant for cardiovascular risk prediction,[6] a prediction model like the one described in our survival analysis was not performed. Our findings suggested the effect of time of the day (from 08:00 hours to 19:00 hours) is not relevant for the CVD risk assessment. With this sensitivity analysis, we wanted to investigate time of day variations beyond simple descriptive diurnal patterns; to our knowledge, this is the first time this finding has been reported.

### Strengths and limitations

The BRHS cohort benefits from using a large-scale population-based sample of free-living older men, and this increases statistical power and precision of estimates. However, the BRHS comprises male participants, predominantly of white European ethnic origin, so findings may not be generalisable to women and non-white ethnic groups. The CVD risk factors measurements were carried out on one occasion over an extended period of the day (between 08:00 and 19:00 hours), offering only a partial understanding of the variations over the 24 hours.[27 28] Therefore, in this study, the relationship of the CVD risk

factors to time of day was explored using between-participant variation only. In future studies, it may be advantageous to carry out repeated measurement of the risk factors over the 24 hours in order to investigate within-person circadian variations. However, with repeated measurements, a possible and genuine diurnal variation may be disrupted and natural sleeping patterns altered (repeated measures are usually taken every 1–2 hours during the night).[29]

### Implications

Variations of some CVD factors (in particular IL-6 and NT-proBNP) over the course of the day were observed, suggesting the role of time of the day as potential confounder during the measurements. However, standardising these biological markers by time of day was not particularly relevant for the cardiovascular risk prediction. Also, other sensitivity analyses (stratified analysis and interaction tests) did not add relevant insights suggesting that time of day variations may be not important for clinical risk stratification in general. Further studies assessing both CVD risk factors levels and clinical outcomes (eg, fatal or non-fatal CVD events) during 24 hours are required to demonstrate whether a rapid increase of IL-6 over the day may be relevant to the increased number of CVD events observed in early and late morning,[30] and whether the increased levels of NT-proBNP over the day are related to the afternoon peak in sudden death following heart failure.[31]

### CONCLUSIONS

Variations in time of day were associated with variations of some, but not all, CVD risk factors measured in older adults. The contribution of time of the day to the markers' overall variation was small and unlikely to affect the CVD risk prediction or clinical risk stratification.

**Contributors** CS processed the data, performed statistical analyses, drafted and revised the manuscript and incorporated revisions of coauthors. PHW, SGW, BJJ and RWM contributed to the design of the study and revised the manuscript. LL enrolled participants and collected data. PHW, RWM and GSW raised grant funding. All authors gave an intellectual contribution to the manuscript and approved the final version.

**Funding** The BRHS is supported by a British Heart Foundation (BHF) programme grant (RG/13/16/30528). This research was supported by a BHF project grant (PG/13/41/30304), which supported CS. The funders had no role in the design and conduct of the study; collection, management, analysis and interpretation of the data; preparation, review or approval of the manuscript; and the decision to submit the manuscript for publication.

**Disclaimer** The views expressed in this publication are those of the author(s) and not necessarily those of the BHF.

**Competing interests** None declared.

**Patient consent** Obtained.

**Ethics approval** The National Research Ethics Service (NRES) Committee for London provided ethical approval.

**Provenance and peer review** Not commissioned; externally peer reviewed.

**Data sharing statement** The collection and management of data over the last 39 years of the BRHS has been made possible through grant funding from UK government agencies and charities. We welcome proposals for collaborative projects and data sharing. For general data sharing enquiries, please contact Lucy Lennon at l.lennon@ucl.ac.uk.

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
