## [Reviewer comments · BMJ Open]

ARTICLE DETAILS

TITLE (PROVISIONAL)	Associations of time of day with cardiovascular disease risk factors measured in older men: results from the British Regional Heart Study
AUTHORS	Sartini, Claudio; Whincup, Peter; Wannamethee, Goya; Jefferis, Barbara; Lennon, Lucy; Lowe, Gordon; Welsh, Paul; Sattar, Naveed; Morris, Richard

VERSION 1 – REVIEW

REVIEWER	Karice Hyun The George Institute for Global Health, Australia
REVIEW RETURNED	28-Jul-2017

GENERAL COMMENTS	This study explores an interesting relationship between time of day and CVD risk factors, and further, CVD risk prediction. The authors used appropriate statistical methods, including the adjustment for the clustering of the town of residence, to explore these relationships. The results are given clearly and the plots give a clear representation of the results. One minor comment would be to spell out the abbreviations under tables and figures.
---

REVIEWER	Per-Olof hansson Department of Molecular and Clinical Medicine, Institute of Medicine, Sahlgrenska Academy, University of Gothenburg, Gothenburg, Sweden
REVIEW RETURNED	07-Aug-2017

GENERAL COMMENTS	This paper have analysed time of day variation of 8 different biomarkers: CRP, fibrinogen, t-PA, von Willenbrant factor, D-dimer, NT-Pro-BNP, Il-6 and Troponin T in an epidemiological study of 4252 men. There is no repeated analysis on each patient; instead the results have been compared for different participants examined at different time of day. Even though I find the paper interesting I have some comments. Major suggestions for revision: 1. I do not really agree on the title that these are “novel cardiovascular disease risk factors”. Fibrinogen was first described to be associated with risk of MI and stroke in 1984. The diagnostics value of D-dimer in venous thromboembolism was first described in 1987. NT pro-BNP have been used I clinical practice for diagnosis of congestive heart failure for several years just as TNT is established for diagnosis of myocardial infarction. Instead a mix of several biomarkers used for different cardiovascular
---

	diseases have been analyzed. But these are not new (or novel) risk factors. 2. There are several questions concerning the methodology: WHEN were the blood samples analyzed and where. The examinations were performed 1998-2000 but was the laboratory analyses performed then or was the blood stored and analyzed later? If so, how? Plasma? Serum? Whole blood? Frozen in minus 70 degrees? There were 24 centers but was analyses performed at all of these centers or at ONE center? In that case where? The blood samples were taken during 11 hours from 08.00 to 19.00. When were the samples transported to the laboratory? Once or twice a day? If the time from blood sample was taken until it arrived to the laboratory varied from less than 1 to 13 hours this might have biased the results. Please clarify the methodology concerning the blood sampling and analysis. 3. Was the examination time of the participants random? In this kind of studies the participants are often offered several times to choose from. In addition patients with some diagnosis such insulin treated diabetes mellitus is often offered a date in the morning in order to minimize the fasting period. Also younger participants (still working) are often offered examination in the late afternoon. What this the case in this study? Such selection might explain the differences described. 4. The differences discovered for 3 biomarkers was small, only significant in trend test (overlapping confidence intervals), very likely without any clinical importance and might be explained by methodological difficulties as suggested above. Therefore my conclusion is that if there is any time of day variation in these biomarkers it is small and probably without any clinical importance. Which is an important clinical message ! 5. You should consider adjusting also for diabetes mellitus in the multivariable model. Does that change the results? Minor suggestions: 6. It is stated that samples are taken between 08:00 and 18:00 but on lines 237 and 272 in the discussion it is stated 08.00 to 18.00. Which is correct? 7. Introduction line 60. It is stated that N-terminal pro-brain natriuretic peptide is a promising marker of heart failure. But it is actually established and widely used in clinical practice. 8. In table 1 there are several abbreviations, such as "HMF". Please write out or explain all abbreviations in the foot note. 9. In table 1 also add number of patients with diabetes mellitus.
--	---

VERSION 1 – AUTHOR RESPONSE

Reviewer: 1

Reviewer Name: Karice Hyun

Institution and Country: The George Institute for Global Health, Australia Please state any competing interests or state 'None declared': None declared

Comment:

This study explores an interesting relationship between time of day and CVD risk factors, and further, CVD risk prediction. The authors used appropriate statistical methods, including the adjustment for the clustering of the town of residence, to explore these relationships. The results are given clearly

and the plots give a clear representation of the results. One minor comment would be to spell out the abbreviations under tables and figures.

Author response:

We thank the reviewer for these positive comments and for the attention to details about tables and figures. We have now spelled out the abbreviations; tables 1 and 2 and figure 1 are now self-explanatory.

Reviewer: 2

Reviewer Name: Per-Olof hansson

Institution and Country: Department of Molecular and Clinical Medicine, Institute of Medicine, Sahlgrenska Academy, University of Gothenburg, Gothenburg, Sweden Please state any competing interests or state 'None declared': None declared

Please see attached file. The methodology concerning the laboratory analysis is incomplete

Comment:

This paper have analysed time of day variation of 8 different biomarkers: CRP, fibrinogen, t-PA, von Willenbrant factor, D-dimer, NT-Pro-BNP, Il-6 and Troponin T in an epidemiological study of 4252 men. There is no repeated analysis on each patient; instead the results have been compared for different participants examined at different time of day.

Even though I find the paper interesting I have some comments.

Major suggestions for revision:

Author response: We thank the reviewer for his interest in our work and we address the reviewer's comments below.

Major suggestions for revision:

Comment 1:

I do not really agree on the title that these are "novel cardiovascular disease risk factors". Fibrinogen was first described to be associated with risk of MI and stroke in 1984. The diagnostics value of D-dimer in venous thromboembolism was first described in 1987. NT pro-BNP have been used I clinical practice for diagnosis of congestive heart failure for several years just as TNT is established for diagnosis of myocardial infarction. Instead a mix of several biomarkers used for different cardiovascular diseases have been analyzed. But these are not new (or novel) risk factors.

Author response:

We thank the reviewer for this comment. We agree to remove "novel" from the title, abstract, and conclusions. We have substituted "novel" with "emerging" in the introduction and discussion. For example, introduction (lines 43-44): "Previous studies have reported time of day variation in both established and emerging cardiovascular disease (CVD) risk factors in middle aged adults". We still think it is relevant to highlight the difference from established risk factors (e.g. lipids) which have been most intensively studied since the 1960s in the aetiology and pathogenesis of CVD.

Comment 2.

There are several questions concerning the methodology:

WHEN were the blood samples analyzed and where. The examinations were performed

1998-2000 but was the laboratory analyses performed then or was the blood stored and analyzed later? If so, how? Plasma? Serum? Whole blood? Frozen in minus 70 degrees? There were 24 centers but was analyses performed at all of these centers or at ONE center? In that case where?

The blood samples were taken during 11 hours from 08.00 to 19.00. When were the samples transported to the laboratory? Once or twice a day? If the time from blood sample was taken until it arrived to the laboratory varied from less than 1 to 13 hours this might have biased the results. Please clarify the methodology concerning the blood sampling and analysis.

Author response:

we agree the methodology concerning the laboratory analysis is incomplete. We have reviewed the methods section and we have strengthened this part specifying the details of the laboratory analysis (lines 116-123). We now confirm that the samples were centrifuged and separated on the morning or afternoon of collection and stored on site at -20°C until they were transferred to a central freezer storage location at -70°C within 2 weeks of sample collection. Samples were then transferred on dry ice to a single central laboratory and were thawed immediately before analysis. Plasma samples were used for all the analyses reported here. The original sample collection took place between January 1998 and March 2000. Most of the analyses described here were carried out during 2000, after a maximum of 3 years storage; NT-ProBNP and cTNT were analysed in 2009. These details have been added to the methodological description.

Comment 3:

Was the examination time of the participants random? In this kind of studies the participants are often offered several times to choose from. In addition patients with some diagnosis such insulin treated diabetes mellitus is often offered a date in the morning in order to minimize the fasting period. Also younger participants (still working) are often offered examination in the late afternoon. What this the case in this study? Such selection might explain the differences described.

Author response:

Participants' appointment times were non-systematically allocated. They were offered the opportunity to contact the BRHS team and change the time of examination, if unable to attend; a small proportion of participants did so. We have now specified these details in the text (see method section lines 81-84). Younger participants were not offered to come at a specific time of the day (e.g. late afternoon). Participants with diabetes (n=478, 11.2%) were instructed not to fast before the examination. Two sensitivity analyses were carried out: (1) models were additionally adjusted for diabetes status; and (2) analysis was performed excluding men with diabetes. Please see our response to point n.5, and new results added in table 2 (Model 3)

Comment 4:

The differences discovered for 3 biomarkers was small, only significant in trend test (overlapping confidence intervals), very likely without any clinical importance and might be explained by methodological difficulties as suggested above. Therefore my conclusion is that if there is any time of day variation in these biomarkers it is small and probably without any clinical importance. Which is an important clinical message !

Author response:

We thank the reviewer for this comment. We agree that variations by time of the day were small, and without clinical importance as our survival analysis demonstrated. We agree this is an important clinical message as reported in our conclusions (see abstract, and conclusion): "The contribution of time of the day to the markers' overall variation was small and unlikely to affect the CVD risk

prediction or clinical risk stratification". We have strengthened this point also in the discussion section (lines 230-234), adding the following statement: "Our analyses showed that the contribution of time of the day to the overall variation of NT-ProBNP, IL-6, and t-PA was small and without clinical importance; we observed that time of day did not have a sufficiently strong effect to be taken into account when assessing the impact of IL-6, NT-ProBNP, and t-PA on CVD mortality".

Comment 5:

You should consider adjusting also for diabetes mellitus in the multivariable model. Does that change the results?

Author response:

We thank the reviewer for this comment. We did consider the role of diabetes. In summary, after adjusting for diabetes status, the estimates reported in Table 2 have not changed. We have now amended the paragraphs "sensitivity analysis" (methods lines 151-152, and results sections lines 192-195), adding a further comment. The sentence is: "An additional adjustment for diabetes status did not alter the magnitude of the association between hour of the day and the outcomes". We also performed the analysis excluding men with diabetes completely and added these results in table 2, model 3. The estimates were very similar to those obtained using the whole sample, and our conclusions remain the same. We now added a statement in the results section "sensitivity analysis". The sentence is: "the exclusion of men with diabetes from the analysis (Table 2 – Model 3) did not substantially change the association between time of day and the outcomes".

Minor suggestions:

Comment 6:

It is stated that samples are taken between 08:00 and 18:00 but on lines and 237 and 272 in the discussion it is stated 08.00 to 18.00. Which is correct?

Author response:

We thank the reviewer for this comment. We have now specified throughout the manuscript that participants were examined between 08:00-19:00h

Comment 7:

Introduction line 60. It is stated that N-terminal pro-brain natriuretic peptide is a promising marker of heart failure. But it is actually established and widely used in clinical practice.

Author response:

We agree the word "promising" can be removed. We edited the sentence accordingly.

Comment 8:

In table 1 there are several abbreviations, such as "HMF". Please write out or explain all abbreviations in the foot note.

Author response:

We would thank the reviewer for pointing this out. We have now spelled out all abbreviations; tables 1 and 2 and figure 1 are now self-explanatory. HMF was substituted with Armed Forces.

Comment 9:

In table 1 also add number of patients with diabetes mellitus.

Author response:

We have now added this information in Table 1; 478 men (11.2%) had diabetes.

VERSION 2 – REVIEW

REVIEWER	Per-Olof hansson Department of Molecular and Clinical Medicine, Institute of Medicine, Sahlgrenska Academy, University of Gothenburg, Gothenburg, Sweden.
REVIEW RETURNED	25-Aug-2017
GENERAL COMMENTS	All my previous questions and comments have been addressed appropriately. I have no further comments

VERSION 2 – AUTHOR RESPONSE

We thank the editor and the reviewers for considering our manuscript suitable for publication in BMJ Open. We have now revised the title of the manuscript including research question, study design and setting. The edited title is:

"Associations of time of day with cardiovascular disease risk factors measured in older men: results from the British Regional Heart Study"